# Real-World Impact of Deep Targeted Sequencing on Erythrocytosis and Thrombocytosis Diagnosis: A Reference Centre Experience

**DOI:** 10.3390/cancers16183149

**Published:** 2024-09-14

**Authors:** Alberto Blanco-Sánchez, Rodrigo Gil-Manso, Rodrigo de Nicolás, Nieves López-Muñoz, Rafael Colmenares, Reyes Mas, Ricardo Sánchez, Inmaculada Rapado, Joaquín Martínez-López, Rosa Ayala Díaz, Gonzalo Carreño-Tarragona

**Affiliations:** 1Centro Nacional de Investigaciones Oncológicas, Complutense University, 28029 Madrid, Spain; al.blanco10@hotmail.com (A.B.-S.); rodrigogm2812@gmail.com (R.G.-M.); rodrigo.denicolas@salud.madrid.org (R.d.N.); marianieves.lopez.munoz@salud.madrid.org (N.L.-M.); rafael.colmenares@salud.madrid.org (R.C.); reyes.masb@gmail.com (R.M.); ricard.sanchez@salud.madrid.org (R.S.); jmarti01@med.ucm.es (J.M.-L.); 2Hematology Department, Hospital Universitario 12 de Octubre, I + 12, 28041 Madrid, Spain; 3Centro de Investigación Biomédica en Red de Oncología, 28029 Madrid, Spain

**Keywords:** erythrocytosis, thrombocytosis, diagnosis, idiopathic, next-generation sequencing

## Abstract

**Simple Summary:**

Around 70% of cases of erythrocytosis are categorised as “idiopathic” after excluding secondary causes and polycythaemia vera. A similar situation arises in the setting of thrombocytosis, with even a 15% of essential thrombocythemia lacking canonical mutations. Previous studies have shown that a deeper investigation of these patients can unmask underlying primary conditions (such as familial disorders or a clonal disease without canonical mutations). The role of next-generation sequencing (NGS) in their diagnosis has been explored in a retrospective manner, showing promising results. In this study, we reviewed the impact of NGS performed in our centre on the diagnosis of erythrocytosis and thrombocytosis (117 and 58 patients, respectively). Our findings showed that few patients benefited from this test, since only 11.9% and 25.9% showed a variant leading to diagnosis of a primary disorder, respectively. However, we believe that this yield could be improved through a better selection prior to NGS.

**Abstract:**

Despite advances in diagnosis of erythrocytosis and thrombocytosis due to driver mutation testing, many cases remain classified as “idiopathic”. This can be explained by the absence of an evident secondary cause, inconclusive bone marrow biopsy or neglection of family history. Analysis of a broad panel of genes through next-generation sequencing (NGS) could improve diagnostic work-up identifying underlying genetic causes. We reviewed the results of NGS performed in our laboratory and its diagnostic impact on 117 patients with unexplained erythrocytosis and 58 with unexplained thrombocytosis; six patients (5.1%) were diagnosed with polycythaemia vera (PV) and 8 (6.8%) with familial erythrocytosis after NGS testing. Low EPO and a family history seemed to predict a positive result, respectively. However, a greater percentage of patients were ultimately diagnosed with secondary erythrocytosis (36%), remained as idiopathic (28.2%) or were self-limited (15%). The yield of NGS was shown to be slightly higher in patients with thrombocytosis, as 15 (25.9%) were diagnosed with essential thrombocythemia (ET) or familial thrombocytosis after variant detection; previous research has shown similar results, but most of them carried out NGS retrospectively, while the present study exhibits the performance of this test in a real-world setting. Overall, the low rate of variant detection and its poor impact on diagnostic work-up highlights the need for a thorough screening prior to NGS, in order to improve its yield.

## 1. Introduction

The aetiologic diagnosis of erythrocytosis or thrombocytosis can be challenging, due to the high prevalence and wide spectrum of secondary causes that can mask an underlying primary disorder [1]. The lack of a gold standard test warrants the conjunction of different criteria prior to the diagnosis of polycythaemia vera (PV) and essential thrombocythemia (ET) [2]. An accurate identification of these disorders is crucial, given the higher risk of thrombosis (amongst others) and the need for specific management [3].

One of the classic criteria, bone marrow (BM) histology [2], is an invasive test, and therefore not suitable for screening. Additionally, morphologic interpretation is not straightforward, as it needs well trained pathologists and it frequently leads to inconclusive results [4].

The assessment of *JAK2*, *CALR* and *MPL* mutational status represented a major step in the diagnosis of PV and ET [5,6,7], and it is currently used as an upfront test as it can be reliably performed on blood samples [1,8]. Their detection provides certainty about the presence of a driver clone, has prognostic value [9] and can even spare BM biopsy [2].

Yet 1–2% and 10–15% of PV [10] and ET [11] cases, respectively, lack driver mutations. In the absence of an evident secondary cause (or when this does not seem to explain the high blood counts), if BM histology is not conclusive, these cases are often categorised as idiopathic [12]. This heterogeneous group could comprise patients with an underlying unidentified cause or even healthy individuals with values above the reference limits. For this reason, some authors argue that cutoffs established by World Health Organisation (WHO) criteria may be too low for PV work-up in the absence of other myeloproliferative features [13]. 

Some cases of *JAK2* wild-type PV could be driven by mutations in other genes [14,15]. Although not included in WHO criteria, some guidelines accept this as a criterion of PV [1]. For ET, the detection of a non-canonical mutation is accepted as a minor criterion by the WHO [2].

Other cases of idiopathic erythrocytosis have proven to be driven by germline mutations of the globin genes or genes involved in the erythropoietin (EPO) or oxygen-sensing signalling pathways [12]. Despite scarce evidence due to their rarity, it is suggested that they carry an augmented risk of thrombosis, and their detection should therefore prompt at least a follow-up [10]. These familial erythrocytosis cases are caused by mutations in genes such as VHL, EPAS, ELGN1 and EPOR [12]. Only the latter has a distinctive trait: low EPO levels. The presence of a family history raises the suspicion of these disorders, which can only be diagnosed after gene sequencing, since they lack specific histologic features [10]. Although even rarer, thrombocytosis is sometimes driven by germline mutations, essentially those affecting *THPO* gene [16] or even in *JAK2* [17] or *MPL* [18].

Next-generation sequencing (NGS) is routinely performed in many myeloid neoplasms at diagnosis, because of its diagnostic and prognostic value [19]. The most common assay is targeted panel sequencing, which allows simultaneous analysis of several candidate genes with enough sensitivity [20]. To date, the best design and its role in the work-up of erythrocytosis and thrombocytosis is not well-stablished [21,22], although the capability of testing various genes makes it an attractive tool. Previous studies have analysed its capacity to unveil genetic conditions in idiopathic populations, showing promising results [11,21,23,24,25,26,27], and some guidelines include NGS in their diagnostic algorithms [1,3,8]. Furthermore, as highlighted in NCCN guidelines, NGS can also provide prognostic information in the setting of PV and ET [8]. 

In the present study, we aimed to analyse in a large cohort of patients the real-world clinical utility of an NGS panel of genes implicated in myeloid neoplasms in the diagnostic work-up of erythrocytosis and thrombocytosis.

## 2. Materials and Methods

### 2.1. Patients and Study Design

Our laboratory performs centralised NGS for most of the haematology departments within Madrid’s region (up to 4 million patients), upon the request of local clinicians. We employ a panel of genes commonly altered in myeloid neoplasms that also includes some of the genes involved in familial erythrocytosis and thrombocytosis. This is a retrospective, multi-centre observational study that included patients diagnosed with idiopathic erythrocytosis or thrombocytosis who underwent NGS requested by their local institutions and performed in our centre between 2016 and 2021. The study was approved by the ethics committee of the Hospital Universitario 12 de Octubre (no. 20/436) and conducted in accordance with the Declaration of Helsinki.

Patients with *JAK2*-negative (as per V617F and exon 12 status) erythrocytosis and no evident or sufficient secondary cause had been categorised as having “idiopathic erythrocytosis”. Likewise, “idiopathic thrombocytosis” had been established after excluding extra haematological causes and ruling out “driver mutations” (*JAK2 V617F*, *MPL W515L* and *CALR* exon 9). 

### 2.2. Next-Generation Sequencing of a Panel of Myeloid Genes

All laboratory tests except NGS had been conducted in the local institution. Laboratory cutoffs were established as per local protocols. Clinical and laboratory data from local centres were collected through the integrated clinical record Madrid network. Variants had been classified according to AMP/ASCO/CAP joint guidelines [28]. Tier I–III variants were reported, although only Tier I and II were considered as clinically relevant. Tier III alterations were reported as variants of uncertain significance (VUS). Details regarding the NGS panel are provided in the Appendix A.

### 2.3. Statistical Analysis

Frequencies were used to describe categorical variables and median and interquartile ranges (IQR) for quantitative variables. A *p* < 0.05 was considered statistically significant. Statistical analysis was conducted using the SPSS computer program version 25.0 (IBM, Chicago, IL, USA).

## 3. Results

### 3.1. Erythrocytosis

Table 1 summarises the main characteristics of the cohort of 117 patients with erythrocytosis. Median age at the time of NGS was 59 years (IQR 43–75). Median time between the first visit and NGS was 14.2 months (IQR 5.8–40.8 months). Only 34 (29.1%) patients had undergone a BM biopsy. NGS was performed in a sample from BM in 29 patients and from peripheral blood in the rest. Twenty-four cases (20.5%) had reported a family history of erythrocytosis, and thus were submitted to NGS under suspicion of familial erythrocytosis. The rest (79.5%) did not show any evident secondary cause, or this did not seem sufficient to induce it. For this reason, they were tested to rule out a clonality marker or a familial erythrocytosis without family history.

Sixty-nine patients (59%) tested positive for genomic variants, but only 20 had variants with clinical utility (Tier I/II), with a median allele frequency of 38%.

Six patients (5.1%) lacking variants were diagnosed as *JAK2*-PV (although only 3 had a conclusive histology). Two showed low EPO, with 3 within the normal range and 1 unknown. Mutations in *JAK2* (all of them in exon 12) were detected in 5 patients and confirmed the diagnosis of PV (3 had a compatible histology and in 2 it was inconsistent, and all of them had low EPO). An additional patient had a mutation in CBL, which, even in the presence of inconclusive histologic findings, led to the diagnosis of PV, although EPO was unavailable. Variant allele frequency (VAF) was under 15% in 3 patients and may be the reason for which it was not detected by Sanger. Two patients showed higher VAF, but Sanger had been performed more than 5 years before NGS, so VAF could have been lower at the initial moment (Appendix A). 

Eight patients tested positive for familial variants and therefore were diagnosed with familial erythrocytosis (4 of them lacking family history, but 2 of these had VUS) (Appendix A). Five additional cases were assumed to have this diagnosis even in the absence of any variant. In 18 patients, the disorder resolved spontaneously, 42 cases were ultimately classified as secondary and 32 remained as “idiopathic”. Figure 1 depicts the percentage of patients established in each of the diagnostic categories after obtaining NGS results. Notably, 9 cases finally diagnosed as secondary or idiopathic harboured variants that were interpreted as clonal haematopoiesis of indeterminate potential (CHIP) (Appendix A).

Patients with reportable variants did not show significant differences in clinical factors such as age or blood counts (Table 2). Low EPO and prior thrombosis seemed to predict the detection of variants leading to a PV diagnosis, and a positive family history was significantly more common in patients with familial erythrocytosis pathogenic variants.

We assessed the contribution of the detection of a significant variant by NGS to the final diagnosis of primary erythrocytosis in terms of sensitivity, specificity, positive predictive value (PPV) and negative predictive value (NPV). Overall, sensitivity was relatively low (0.54; 95% CI 0.35–0.73), while specificity (0.95; 95% CI 0.9–0.99) and NPV (0.88; CI 0.84–0.92) were high, with a modest PPV (0.74; 95% CI 0.54–0.93).

### 3.2. Thrombocytosis

Table 3 summarises the main characteristics of the 58 patients in this cohort. Median age at the time of NGS was 54.5 years (46.6–62.4). Median time between the first visit and NGS request was 9.8 months (IQR 4.2–32.6 months). Most of the patients (46, 79.3%) had undergone a BM biopsy. NGS was performed in a sample from BM in 30 patients and from peripheral blood in the rest. Presence of family history was not systematically recorded.

Thirty-one patients (53.4%) tested positive for variants, and 15 (25.9%) presented with variants with clinical utility (Figure 2). This led to diagnosis of familial thrombocytosis in two of them and ET in thirteen (together with other criteria). One individual showed a deletion in *CALR* exon 9, not detected previously by PCR. Interestingly, three patients harboured mutations in *JAK2*, concretely in exons 12, 14 and 20 (Appendix A). Two of them raised concern about the possibility of familial ET due to VAF near 50%. It was confirmed to be a familial disorder in one of them, after diagnosing four relatives with thrombocytosis and the same mutation.

Twenty-one patients (36.2%) were diagnosed with ET even in the absence of a significant variant but meeting other WHO criteria. In seven patients (12.1%) the disorder resolved spontaneously, nine cases (15.5%) were ultimately classified as secondary and six (10.3%) remained as “idiopathic”. None of the latter showed variants leading to diagnosis of CHIP. 

Patients with reportable variants did not show significant differences in clinical factors such as age, prior thrombosis or blood counts (Table 4). A consistent histology may show a trend towards detection of significant variant, but this did not reach statistical significance.

Regarding the predictive analysis, NGS showed a high specificity and PPV (100%), since all patients with a reportable variant were diagnosed with either ET or familial thrombocytosis. However, sensitivity (0.42; 95% CI 0.26–0.58) and NPV (0.51; CI 0.51–0.51) were low. 

## 4. Discussion

This study evaluates the impact of NGS on the diagnosis of patients with unexplained erythrocytosis or thrombocytosis. Previous studies have outlined its utility in this setting [11,21,23,24,25,26,27], but most of them in a retrospective manner. In our research, we assessed the role of this test in a real-world context, requested by clinical haematologists and integrated in the diagnostic work-up.

Molecular testing for driver mutations is often the most straightforward approach to differentiating between primary and secondary erythrocytosis or thrombocytosis [23]. *JAK2 V617F* PCR is recommended as an upfront test for its reliability, in contrast to the sensitivity of EPO or other clinical factors [29]. Despite its generalised analysis (plus exon12, *CALR* and *MPL*), many cases remain unexplained (up to 70% amongst those with erythrocytosis) [12,22]. For this reason, some studies have aimed at uncovering the genetic origin of these idiopathic disorders through sequencing of candidate genes, with relative success [11,21,23,24,25,26,27]. The frequence of inconclusive BM reports and the uncertainty about other putative but insufficient causes may push clinicians to use NGS as a powerful tool to confirm or rule out a genetic cause, either inherited or acquired.

Nevertheless, not only is this test costly but also it can generate more uncertainty. VUS should be interpreted with caution [28], and germline variants with predisposition to cancer or CHIP can arise as unexpected findings. These are not routinely studied, but once detected, they can result in considerable psychological impact [30] and may prompt a follow-up of otherwise healthy individuals [31,32].

The prevalence of reportable gene variants in our erythrocytosis cohort is low, and only 11.9% of the patients could be classified as having PV or familial erythrocytosis after NGS. This rate is similar to previous studies [21,23,24,25,26]. Applying a myeloid panel, Bhai et al. [23] reported 6% Tier I–II variants in their cohort. With a mixed panel (myeloid and familial), Benetti et al. [24] found 4.2%, although they uncovered the HFE H63D variant in two-thirds of patients, raising the question of a possible link between iron metabolism gene variants and erythrocytosis.

Camps et al. [21] designed a panel for familial erythrocytosis with new candidate genes, allowing for definitive genetic diagnosis in 7.2% of 135 patients. Yet candidate variants were detected in 29%, highlighting the need for further research in the field of familial erythrocytosis. Tier I–II variants were reported by Lee et al. [26] in 9.1% of patients and in 5.9% by Jalowiec et al. [25].

In our study, a similar percentage of patients were diagnosed with PV despite the lack of NGS findings. This fact should discourage NGS testing if other criteria are clearly met, especially BM histology. However, as previously demonstrated [33], it could “rescue” the diagnosis of exon 12 mutated PV with low VAF, on account of Sanger sensitivity and false-negative rate. Accordingly, NGS may be appropriate in the context of a dubious histology, or at least not requested unless a biopsy has been performed. Two of our patients had a considerable VAF (Appendix A), but Sanger had been performed more than 5 years before NGS, when allele burden might have been lower. Repeating Sanger could therefore be an option when NGS is not available. 

Our most remarkable finding is that a high percentage of patients (36%) were eventually diagnosed with a secondary erythrocytosis. This suggests that the main outcome of NGS was reconsidering the secondary source of the disorder and reassuring the clinician about it. Moreover, in self-limited cases (15%), molecular testing could have been avoided.

We hence tried to identify which factors could help select individuals with greater probability of harbouring reportable variants. Some authors have suggested raising the cutoff for further investigation of *JAK2*-negative erythrocytosis (strict criteria), restricting wide criteria (WHO criteria) to PV diagnosis [34]. We did not find significant differences between haemoglobin and haematocrit, although these values were collected only at the time of referral to haematology.

The two factors that appeared to predict the detection of a reportable variant were low EPO (for PV and EPOR mutation) and a positive family history (for familial erythrocytosis). In fact, the patient with EPOR mutation had been followed for two years for secondary erythrocytosis (attributed to sleep apnoea syndrome). He underwent NGS because of his family history and sustained low EPO.

Based on the foregoing, we propose an algorithm for the optimisation of NGS in the setting of *JAK2*-negative erythrocytosis (Figure 3).

Regarding the thrombocytosis cohort, NGS showed a higher yield compared to erythrocytosis, with 25.9% of patients being diagnosed with a primary thrombocytosis after molecular testing. This is in line with a similar study in a TN population, demonstrating mutations in 29.2% [35], although most of the studies have analysed patients with an established diagnosis of ET (with a yield of 35–52.4%) [22,36].

The research by Michail et al. [37] showed poor results: only three out of twenty-seven TN patients harboured mutations. Interestingly, eight individuals from the original cohort did not undergo NGS because of detection of canonical mutations after new molecular screening. The lower yield could be explained by the following: only 66.6% had an available biopsy, and only 25% had a consistent histology.

Patients with thrombocytosis are more prone to undergo BM biopsy, mainly because of concern about myelofibrosis [38]. Additionally, causes of secondary thrombocytosis may be easier to rule out or correct, such as iron deficiency, infection or surgery [39]. Lastly, the rate of TN ET is significantly higher than that of *JAK2*-PV. These factors may promote proper selection, ensuring higher yield of molecular testing. A suggested algorithm for the use of NGS in this setting is shown in Figure 4.

Four patients showed pathogenic variants with low VAF (≤5%) in *TET2*, *DNMT3A* or *TP53*. The interpretation of these variants may be challenging, as previously noted [35], since they could constitute CHIP and not truly a driver event of ET. Caution should be kept especially in the case of TP53, a typically late-occurring event [40]. In fact, all individuals with a significant variant were diagnosed with primary thrombocytosis, leading to a specificity and PPV of 100%, in contrast to the erythrocytosis cohort. In the latter, some patients with pathogenic variants did not meet other criteria of PV, and this disorder was ruled out. Finally, some of the cases with thrombocytosis and a reportable variant had a normal or inconclusive histology and may have been erroneously classified as ET (Appendix A), stressing the importance of an integrated diagnosis in these diseases.

This study has some limitations and strengths. First, there were no standard thresholds or definitions among participant centres, and BM samples were not reviewed. Thus, both cohorts were heterogenous in many aspects. Secondly, we did not collect prospective data nor review VUS, which could have led to reclassification. 

However, our study gives insight into the use of NGS in a real-world setting, emphasising the need for an optimal selection of patients. We suggest a work-up algorithm for patients with idiopathic erythrocytosis, the population in which, in our opinion, the role of NGS is more controversial.

## 5. Conclusions

NGS is a powerful tool for the diagnosis of both *JAK2*-negative erythrocytosis and TN ET. However, it should be cautiously incorporated in the diagnostic work-up of these patients, on account of its high cost, low yield in unselected populations and the potential incidental findings. This may be warranted by a thorough investigation of alternative causes and family history. Further research on the best panel design and interpretation of new candidate variants are needed.

## Figures and Tables

**Figure 1 cancers-16-03149-f001:**
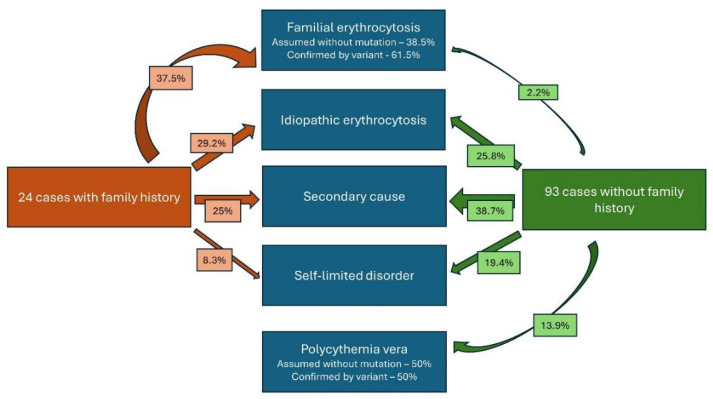
Categorisation of patients with previous idiopathic erythrocytosis after integration of NGS results.

**Figure 2 cancers-16-03149-f002:**
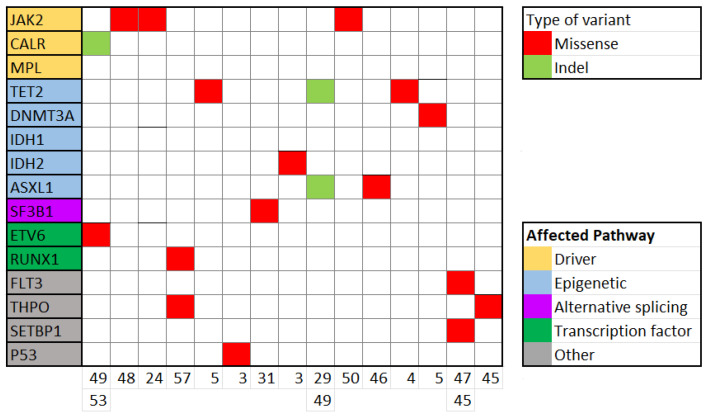
Mutational landscape of patients with thrombocytosis and significant variants.

**Figure 3 cancers-16-03149-f003:**
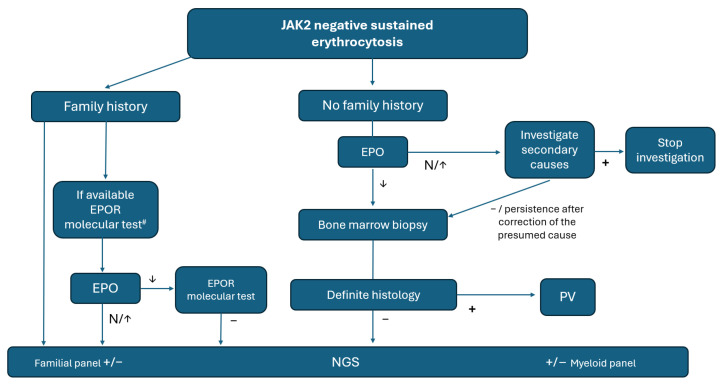
Proposed algorithm for work-up of idiopathic erythrocytosis. ^#^ Laboratories with targeted EPOR analysis could avoid NGS testing.

**Figure 4 cancers-16-03149-f004:**
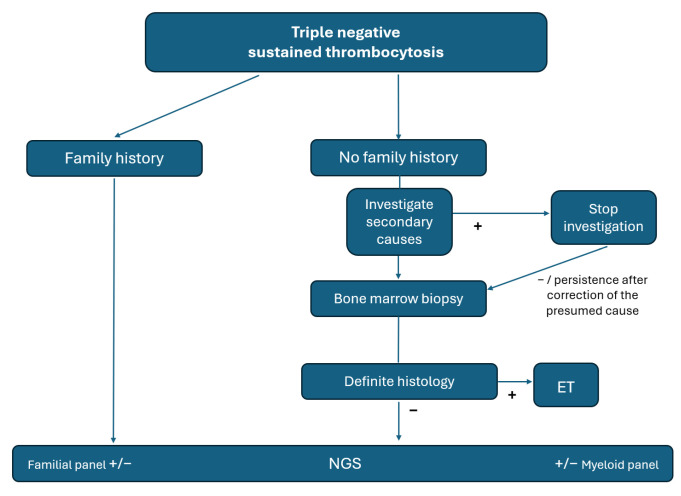
Proposed algorithm for work-up of idiopathic thrombocytosis.

**Table 1 cancers-16-03149-t001:** Baseline characteristics of the erythrocytosis cohort.

Clinical Characteristics (*n* = 117)	
Age at the time of NGS	59 (43–75)
Male (*n*, %)	87 (74.4%)
Female (*n*, %)	30 (25.6%)
Previous thrombosis (*n*, %)	13 (11.1%)
Complete blood count (at first visit)	
Haemoglobin (g/dL)	17.6 (17.1–18.1)
Haematocrit (%)	53.1 (51.6–54.6)
Leucocytes (×10^9^/L)	6.9 (5.8–8)
Platelets (×10^9^/L)	225 (187–263)
Other objective parameters	
Normal or elevated EPO (*n*, %)	98 (83.8%)
Low EPO (*n*, %)	15 (12.8%)
Not available (*n*, %)	4 (3.4%)
Bone marrow available (*n*, %)	34 (29.1%)
Bone marrow consistent with MPN (*n*, %)	6 (17.6%)
Bone marrow inconclusive (*n*, %)	14 (41.2%)
Normal histology, excluding PV (*n*, %)	14 (41.2%)
Abdominal ultrasonography available (*n*, %)	100 (85.5%)
No splenomegaly (*n*, %)	100 (100%)

**Table 2 cancers-16-03149-t002:** Clinical and analytical characteristics of the erythrocytosis cohort according to the presence of any diagnostic variant, a PV-defining variant or a familial erythrocytosis variant.

	No Mutation (*n* = 103)	Diagnostic Variant (*n* = 14)	*p*	PV Variant (*n* = 6)	*p*	Familial Variant (*n* = 8)	*p*
Male, %	78.6%	71.4%	0.7	83.3%	1	62.5%	0.6
Age at first visit (years)	49.9 ± 7	58.7 ± 9	0.1	59.6 ± 22	0.2	56.7 ± 15.9	0.4
Hb at first visit (g/dL)	17.5 ± 0.6	17.6 ± 0.4	0.6	17.5 ± 0.5	0.6	17.6 ± 1	0.9
Hct at first visit (%)	52.9 ± 2.4	53.4 ± 1.9	0.7	54.7 ± 4.2	0.3	52.4 ± 3.6	0.6
Leucocytes (×10^9^/L)	7.3 ± 0.8	6.7 ± 0.5	0.3	6.2 ± 0.9	0.2	7.3 ± 0.7	0.9
Platelets (×10^9^/L)	223 ± 18.5	234 ± 24	0.3	225 ± 71	0.7	242 ± 33	0.3
Previous thrombosis	7.8%	28.6%	0.015	50%	0.01	12.5%	0.4
Low EPO	9.4%			83.3%	0.01		
Family history	19.4%					50%	0.65
						66.6%	0.02 *

Data are given as median (interquartile range), or absolute or relative frequency (percentage). * Family history becomes significant when excluding two patients with VUS. *p* refers to comparison between the no-mutation group and each of the other groups.

**Table 3 cancers-16-03149-t003:** Baseline characteristics of the thrombocytosis cohort.

Clinical Characteristics (*n* = 58)	
Age at the time of NGS	54.5 (46.6–62.4)
Male (*n*, %)	12 (20.7%)
Female (*n*, %)	46 (79.3%)
Previous thrombosis (*n*, %)	4 (6.9%)
Complete blood count (at first visit)	
Haemoglobin (g/dL)	14 (13.4–14.6)
Haematocrit (%)	41.8 (39.5–44.2)
Leucocytes (×10^9^/L)	8.9 (7.4–10.4)
Platelets (×10^9^/L)	618 (523–713)
Other objective parameters	
Bone marrow available	46 (79.3%)
Bone marrow consistent	16 (34.8%)
Bone marrow inconclusive	18 (39.1%)
Bone marrow normal	12 (26.1%)
Abdominal ultrasonography available	40 (69%)
No splenomegaly	100 (100%)

**Table 4 cancers-16-03149-t004:** Clinical and analytical characteristics of the thrombocytosis cohort according to the presence of a diagnostic variant.

	No Mutation (*n* = 43)	Diagnostic Variant (*n* = 15)	*p*
Female	82.9%	70.6%	0.3
Age at the time of NGS (y)	56 ± 11.5	59 ± 13	
Platelets at first visit	575 ± 115	725 ± 200	0.8
Hb at first visit (g/dL)	13.9 ± 0.8	14.1 ± 1.1	0.9
Hct at first visit (%)	41.2 ± 3	44.3 ± 2.3	0.6
Leucocytes at first visit (×10^9^/L)	8.3 ± 1.3	8.2 ± 3.1	0.5
Thrombosis at first visit (×10^9^/L)	4.9%	11.8%	0.3
Bone marrow consistent	28.1%	40%	
Bone marrow inconclusive	43.8%	40%	
Bone marrow normal	28.1%	20%	0.3

## Data Availability

The raw data supporting the conclusion of this article will be made available by the authors, without undue reservation.

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
