# Peer review of "Real-World Impact of Deep Targeted Sequencing on Erythrocytosis and Thrombocytosis Diagnosis: A Reference Centre Experience"

_cancers, 2024, doi:10.3390/cancers16183149_

Round 1

Reviewer 1 Report

Comments and Suggestions for Authors

The article is of interest to hematology readers as it opens the horizon for using a more extensive NGS panel to diagnose less straightforward cases of polycythemia and thrombocytosis. The Proposed algorithm for the workup of idiopathic erythrocytosis could be an addition to clinical practice. The article, however, is difficult to read as tables 1 and 3 have redundant information like values at NGS, Maximum values and values at first visit. In contrast, one parameter (at NGS, for example) would be sufficient to reflect the abnormality. The authors should also refer to the newer NCCN Clinical Practice Guidelines in Oncology (NCCN Guidelines®) Myeloproliferative Neoplasms Version 3.2023 — October 25, 2023, which uses the NGS as an essential tool to stratify patients and deliver a precision treatment strategy.

Author Response

Thank you very much for taking the time to review this manuscript. Please find the detailed response below and the corresponding revisions highlighted changes in the re-submitted file.

Comment: The article is of interest to hematology readers as it opens the horizon for using a more extensive NGS panel to diagnose less straightforward cases of polycythemia and thrombocytosis. The Proposed algorithm for the workup of idiopathic erythrocytosis could be an addition to clinical practice. The article, however, is difficult to read as tables 1 and 3 have redundant information like values at NGS, Maximum values and values at first visit. In contrast, one parameter (at NGS, for example) would be sufficient to reflect the abnormality. The authors should also refer to the newer NCCN Clinical Practice Guidelines in Oncology (NCCN Guidelines®) Myeloproliferative Neoplasms Version 3.2023 — October 25, 2023, which uses the NGS as an essential tool to stratify patients and deliver a precision treatment strategy.

Response: we agree with this comment. Hemoglobin and platelet values at one time of follow-up are sufficiently explanatory, while the other values do not add on any further information. Therefore, we have deleted this extra data from tables 1-4. We think this will make the manuscript clearer.

Regarding NCCN Clinical Practice Guidelines, they indeed highlight the value of NGS in the diagnostic work-up of erythrocytosis and thrombocytosis lacking canonical mutations (page 8 of NCCN Guidelines on Myeloproliferative Neoplasms). Consequently, we refer to these guidelines in the re-submitted file (reference 8). Additionally, NCCN guidelines underline the prognostic value of other mutations detected by NGS, so we have modified the manuscript to mention this key feature (page 2, paragraph 6, lines 86 and 87).

Reviewer 2 Report

Comments and Suggestions for Authors

The authors focus on a relevant clinical issue, i.e. the correct diagnosis of erythrocytosis and thrombocytosis lacking the classical driver mutations.

They provide a thorough analysis of their real-life data and suggest a valuable clinical approach, to try and solve a real challenge for clinicians in the daily life setting.

My suggestion is to add a figure for the proposed algorithm for work-up of idiopathic thombocytosis (as the one provided for idiopatic erythrocytosis).

Author Response

Thank you very much for taking the time to review this manuscript. Please find the detailed response below and the corresponding revision in the re-submitted file.

Comment 1: The authors focus on a relevant clinical issue, i.e. the correct diagnosis of erythrocytosis and thrombocytosis lacking the classical driver mutations.

They provide a thorough analysis of their real-life data and suggest a valuable clinical approach, to try and solve a real challenge for clinicians in the daily life setting.

My suggestion is to add a figure for the proposed algorithm for work-up of idiopathic thombocytosis (as the one provided for idiopatic erythrocytosis).

Response 2: we agree with this comment. Therefore, we have added a new figure in the manuscript with an algorithm for the work-up of idiopathic thrombocytosis. We think this can help clinicians guide their diagnostic work-up and use of NGS in daily practice. This modification can be seen on page 9, line 294.